# Dosage compensation of Z sex chromosome genes in avian fibroblast cells

Ruslan Deviatiiarov[1,2,3,4†], Hiroki Nagai[1†], Galym Ismagulov[1], Anastasia Stupina[2], Kazuhiro Wada[5], Shinji Ide[6], Noriyuki Toji[5], Heng Zhang[7], Woranop Sukparangsi[8], Sittipon Intarapat[9], Oleg Gusev[2,3,4*] and Guojun Sheng[1*]

---

[†]Ruslan Deviatiiarov and Hiroki Nagai contributed equally to this work.

*Correspondence:
o.gusev.fo@juntendo.ac.jp;
sheng@kumamoto-u.ac.jp

[1] International Research Center for Medical Sciences, Kumamoto University, Kumamoto, Japan
[2] Regulatory Genomics Research Center, Institute of Fundamental Medicine and Biology, Kazan Federal University, Kazan, Russian Federation
[3] Graduate School of Medicine, Juntendo University, Tokyo, Japan
[4] Life Improvement by Future Technologies Institute, Moscow, Russian Federation
[5] Department of Biological Sciences, Faculty of Science, Hokkaido University, Sapporo, Japan
[6] Kumamoto City Zoo and Botanical Garden, Kumamoto, Japan
[7] Graduate School of Life Science, Hokkaido University, Sapporo, Japan
[8] Department of Biology, Faculty of Science, Burapha University, Chonburi, Thailand
[9] Faculty of Science, Mahidol University, Bangkok, Thailand

## Abstract

In birds, sex is genetically determined; however, the molecular mechanism is not well-understood. The avian Z sex chromosome (chrZ) lacks whole chromosome inactivation, in contrast to the mammalian chrX. To investigate chrZ dosage compensation and its role in sex specification, we use a highly quantitative method and analyze transcriptional activities of male and female fibroblast cells from seven bird species. Our data indicate that three fourths of chrZ genes are strictly compensated across Aves, similar to mammalian chrX. We also present a complete list of non-compensated chrZ genes and identify Ribosomal Protein S6 (RPS6) as a conserved sex-dimorphic gene in birds.

**Keywords:** Birds, Sex chromosome, Dosage compensation, Sex determination, Fibroblast, CAGE (cap analysis of gene expression), TSS (transcription start site), CASI (cell autonomous sex identity), RPS6 (ribosomal protein S6)

## Background

The molecular mechanisms regulating avian sex determination are not well-understood. Sex in birds is specified genetically, with the female being heterogametic (ZW sex chromosomes) and male homogametic (ZZ sex chromosomes) (Fig. 1A) [1–4], opposite to eutherian mammals (male XY and female XX). Phylogenomic analyses indicated that Z and X sex chromosomes (chrZ and chrX) evolved separately from different autosomes and likely originated from dimorphic expression of autosomal genes involved in sex differentiation [5]. Suppression of meiotic recombination of chrZ/chrX led to their degeneration into chrW/chrY, respectively [6]. In many non-avian/non-mammalian amniotes (all crocodiles, all sphenodons, some squamates and turtles), sex determination is governed not genetically but rather via environmental factors (e.g., temperature) during critical developmental windows (Fig. 1A) [7]. It has been hypothesized that temperature sex determination was the ancestral mode in amniotes [7].

A well-studied example of dosage-compensation is XIST (a long non-coding RNA)-mediated chrX condensation and subsequent inactivation in placental mammals [8–10].

**Fig. 1** Dosage compensated and non-compensated genes in male and female fibroblast cells in birds. **A** Amniote phylogeny and sex determination. **B** Numbers of active TSS (transcription start sites) and genes in avian fibroblast cells. **C** Numbers of differentially expressed genes between male and female fibroblast cells in chicken. **D** MDS (multi-dimensional scaling) plot for chicken fibroblast samples. Two groups of samples are clearly separated by dimension 1. CAGE-seq samples labeled by "C07-12," RNA-seq samples by "male" and "female". **E** Hierarchical clustering of sex chromosome gene expression determines fibroblast sex identity in chicken. Male samples lack chrW gene expression (arrow). **F** Expression profiles of known sex markers in chicken (C11 and C12: male; C7-10: females). FDR values are shown. **G** Expression fold changes (FC) of genes located on chicken sex chromosomes and autosomes. Male boxplot for genes with log2FC > 0, female log2FC < 0 (FDR < 0.05). Genes localized on chrW demonstrated the greatest differences, while chrZ genes have relatively narrow FC ranges between male and female. Wilcoxon test was used for statistical estimates. **H** Numbers of differentially expressed (DE) genes between male and female by chicken chromosomes. Arrows indicate sex chromosomes. **I** Percentage of compensated genes (FDR > 0.05) on chrZ in birds. On average 78% of genes show no DE between sexes in fibroblast cells (with no FC cutoff, excluding emu and peacock). For emu, compensation ratio is 29% when PAR is excluded, blue bars indicate percentage of non-DE genes on entire chrZ. Peacock excluded due to absence of chrZ in its genome assembly

In birds, however, chrZ does not undergo whole-chromosome condensation or transcriptional inactivation [11]. Estimate of percentage of compensated chrZ genes varies in literature (~20–45%, [12–14]). It has been proposed that avian dosage compensation is regulated locally and that lack of compensation of chrZ genes (e.g., *DMRT1*) is necessary to drive sex-specific germ cell and gonadal differentiation [15–20]. In addition to

dimorphism in gonadal morphology, birds exhibit secondary sexual dimorphism (e.g., body weight, plumage shape, wattle size and courtship behavior) which is mainly under cell-autonomous regulation via inherited genetic identity and weakly modulated by sex hormones after gonadal differentiation [19, 21–23].

## Results and discussion

To investigate how sex-specific differentiation in birds is programed cell-autonomously, we compared transcriptional activities in purified male and female embryonic fibroblast cells from seven avian species (the chicken, quail, turkey, blue peafowl, duck, zebra finch and emu), representing all three major clades: the Palaeognathae (emu), Neoaves (zebra finch), and Galloanserae (the rest) (Materials and Methods) (Fig. 1A). Sex was assessed by gonad size [4], and in each species, at least three embryos (including both male and female) were collected for fibroblast isolation (one embryo per sample). Two different culture conditions (DMEM high glucose and DMEM/F12) (Materials and Methods) were used to control medium-associated expression variation. Cultured cells were passaged five times to reach homogeneity for RNA isolation (Additional file 1: Fig. S1A, showing chicken, peafowl and emu fibroblasts). All cells grew robustly, and no prominent difference was observed between fibroblast cells under different culture conditions or from different origins (individuals, sexes or species), suggesting that our fibroblast cell isolation method and culture protocol are applicable to all birds.

Cap analysis of gene expression (CAGE) maps gene expression by capturing and sequencing 5′ end of mRNA [24, 25]. CAGE-seq can identify transcription start sites (TSSs) with single nucleotide precision and quantify TSS activities free of amplification- or transcript size-introduced biases [26, 27], ideal for analysis of minor expression variations between male and female cells. Applicability of CAGE-seq to chicken samples was reported previously [28]. We decided to use CAGE-seq to investigate sex-biased gene expression in male and female fibroblast cells from these seven species (Fig. 1A; Additional file 1: Fig. S1B; Additional file 2: Table S1; Materials and Methods). CAGE replicates correlated well with each other and with corresponding RNA-seq data used for validation (Additional file 1: Fig. S2, S3).

On average, 8582 TSS peaks representing 7608 genes (Fig. 1B; Additional file 1: Fig. S4; Additional file 3: Table S2) were identified in avian fibroblast cells. Of those, 95.4% (7269 genes) did not show statistically significant difference between the male and female samples (Fig. 1C; Additional file 4: Table S3), and 4.5% (339 genes) showed either male-biased (189) or female-biased (150) differential (sex dimorphic) expression (Fig. 1C; Additional file 4: Table S3). Dimorphism in sex-associated TSS activities had the strongest effect on sample clustering (Fig. 1D; Additional file 1: Fig. S5A), largely due to differential activities of both chrW and chrZ genes in all species studied except for the emu (Fig. 1E; Additional file 1: Fig. S5B, Additional file 5: Table S4) and in agreement with several reported chrW markers (e.g., *HINTW*, *CHDB1*, and *ATP5A1W*) (Fig. 1F) [29]. Although differentially expressed genes were significantly enriched on sex chromosomes (Additional file 6: Table S5), many of them were also located on autosomes (1736 out of 2370, 73.2%; Additional file 4: Table S3), and these genes had an average male/female ratio higher than chrZ genes (Fig. 1G), suggesting that dimorphic expression of chrZ/chrW genes likely control sex specification through autosome genes.

The number of dimorphic genes located on sex chromosomes (Z and W) in each species is shown in Additional file 4: Table S3. In chicken, of the 132 sex chromosome genes with sex specific differential expression, 115 were on the chrZ and 17 on chrW (Fig. 1H; Additional file 4: Table S3). This indicated that of the 387 chrZ genes that were expressed in fibroblast cells, only 30% (115/387) exhibited statistically significant difference between the two sexes, with the rest showing no significant difference. In other birds, the percentages of expressed genes showing statistically significant difference were 6% (23/392) in quail, 35% (123/356) in turkey, 22% (85/380) in duck, 25% (101/411) in zebra finch, and 19% (63/324) in emu (Additional file 4: Table S3). When we excluded peafowl sex chromosome data (due to poor chrZ annotation) [30], number of genes on chrZ showing no statistical difference between the male and female samples were consistently high in the remaining species (chicken 70%; quail 94%; turkey 65%; duck 78%; zebra finch 75%; and emu 79% [29% when excluding PAR]) (Fig. 1I). These percentages increased further when we applied male/female expression ratio cutoffs (e.g., 1.3-fold and 1.5-fold). Using the most stringent criterion for non-compensation (i.e., genes exhibiting any statistically significant difference between the sexes without applying any ratio cutoff) and not considering PAR in Neognathae species, the average percentage of dosage-compensated chrZ genes was 78% (Fig. 1I). This number is significantly greater than previous estimates for avian chrZ dosage compensation (see introduction) and is comparable with the percentage of X-linked, dosage-compensated genes reported in human fibroblasts [~75% in [31] and ~70% in [32]], suggesting that avian dosage compensation works as effectively as in mammals despite the lack of morphologically-distinct chrZ inactivation in birds. Contribution of gametologous genes to differential expression is available in Additional file 7: Table S6, and examples of expression profile are shown in Additional file 1: Fig. S6.

How birds modulate their chrZ gene expression in male versus female cells is unclear. In addition to chromosome-level inactivation in homogametic sex (e.g., female mammals, XX), dosage compensation can be achieved through doubling sex chromosome expression in heterogametic sex (e.g., male *Drosophila*, XY) [33], halving sex chromosome expression in homogametic sex (e.g., female nematodes with XX; and male silkworms with ZZ) [34, 35], or a hybrid mechanism (e.g., in monarch butterfly, halving expression in one chrZ segment in homogametic sex and doubling expression in another chrZ segment in heterogametic female) [36]. Our data indicated obvious regionalized distribution of either compensated or non-compensated genes on avian chrZ in the case of emu, in which male-biased chrZ genes appeared to cluster to the stratum 0, whereas PAR (pseudo-autosomal region)-located genes, as expected, did not show sex-specific expression shifts (Additional file 1: Fig. S7A). Stratum 0 and PAR are previously reported features of homomorphic sex chromosome evolution in birds [37, 38]. Autocorrelation analysis revealed weak regionalization of non-compensated chrZ genes in several non-ratite species in comparison to the emu (Additional file 1: Fig. S8).

Sex dimorphic gene expression in birds is known to appear before gonadal differentiation [21, 22] and continue afterwards. However, *DMRT1*, a chrZ gene and a putative determining factor in testis differentiation, is not expressed in fibroblast cells and does not regulate secondary sex dimorphic characters [19, 20]. Such gonadal factor-independent sex-specific somatic cell specialization is referred to as cell autonomous sex

identity (CASI) [23, 39], molecular mechanism of which is unknown. We analyzed our datasets for differentially expressed genes between male and female fibroblast cells in each species (Additional file 8: Table S7 with top 50 highlighted; top 8 shown in Additional file 1: Fig. S9). We noticed four interesting features, both validating previous findings and offering new insight to dosage compensation and CASI. First, one chicken chrZ gene (LOC112530614) exhibited strong inactivation in male fibroblast cells. This gene is located within the *MHM1* (male hypermethylation 1) locus (Additional file 1: Fig. S7B) reported to be downregulated in male and active in female [14, 17, 40] and hypothesized to positively regulate female chrZ genes [41]. Although this *MHM1* locus is conserved in Galloanserae, it is not universally present in birds [14, 42]. Second, *HINTW* was listed as a chrW gene expressed in a female-specific manner in the chicken (Fig. 1F), validating previous reports from our and other labs [21, 43–45]. Direct- and cross-mapping suggested presence of female-specific *HINTW* expression in all species analyzed except for the emu (in which the homomorphic nature of its chrZ and chrW precluded us from making an unambiguous prediction). Third, many genes found in this study had not been reported previously and constituted a valuable resource for future functional studies of CASI in each species (Additional file 8: Table S7) (e.g., chicken *COL4A1* and *AHCY*, quail *HSPA8*, and emu *RAD23*). Lastly, we uncovered a gene that was highly expressed and universally sex-dimorphic in all seven species (Additional file 1: Fig. S10A). This gene is located on chrZ and encodes *RPS6* (ribosomal protein subunit 6) (Fig. 2A). It is

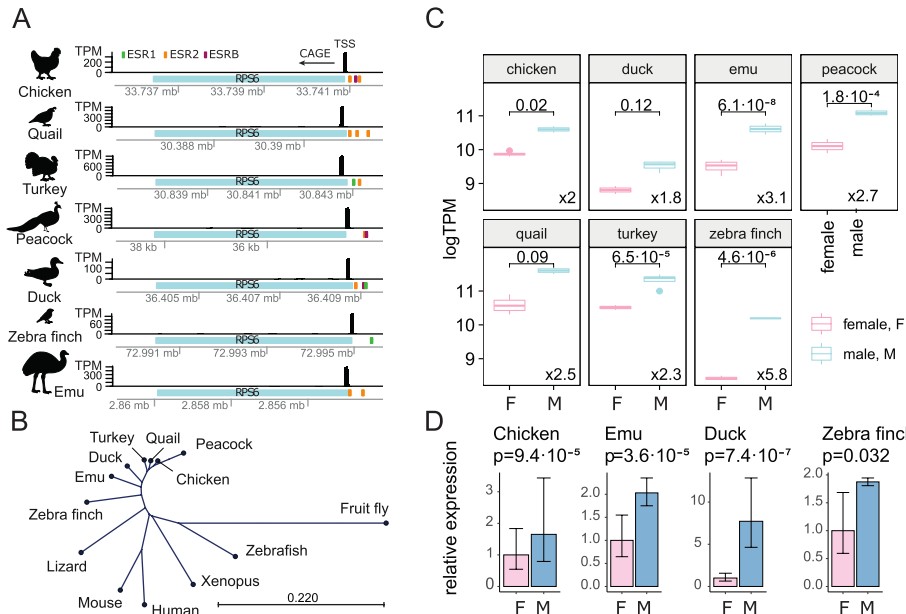

**Fig. 2** RPS6 is a universal marker of sex identity in avian fibroblast cells. **A** Genomic view of *RPS6* in all studied species. This gene has single sharp TSS peak, one annotated isoform, and similar sizes in all species. Promoter sequences (500 bp) were analyzed for presence of estrogen hormone receptor DNA binding motifs. Motif matches with *p* value < 0.001 are shown. **B** A phylogenetic tree for *RPS6* based on nucleotide sequences alignment. **C** CAGE expression profiles of *RPS6* gene. Expression of RPS6 is higher in male fibroblast cells. FDR values from differential expression analysis are shown. Bottom right numbers represent fold change between male and female samples. **D** qPCR for *RPS6* confirmed statistically stable up-regulation of the gene in male fibroblast cells. Number of biological replicates for chicken—3 males, 4 females (× 2 technical replicates). For other species, number of replicates is same in CAGE experimental design

extremely conserved evolutionarily (Fig. 2B, Additional file 1: Fig. S10B). Stably higher *RPS6* expression in male cells at stages prior to gonadal differentiation (Additional file 1: Fig. S11A) and male-biased expression in all tested avian fibroblast cells indicated by CAGE (Fig. 2C) suggest its universal role in CASI. Differential expression of *RPS6* in male and female fibroblast cells was further validated by quantitative PCR using fibroblast RNA from the chicken, quail, duck, and zebra finch (Fig. 2D). Furthermore, *RPS6* expression was not affected by tested media conditions (Additional file 9: Table S8), although potential culture media-introduced bias in fibroblast transcriptome cannot be ruled out and awaits further investigation (Additional file 1: Fig. S12).

*RPS6* has been implicated in metabolic regulation in mouse embryonic fibroblast cells [46] and cell cycle regulation in chicken embryonic fibroblast cells [47], suggesting its involvement in sex-specific metabolic and/or cell-cycle regulation. Supporting a potential role of *RPS6* in avian CASI, this gene was reported to play a role in sex-specific vocal learning in zebra finch [48], and its expression exhibited both cell-autonomous dimorphism and estrogen inducibility in song regions of the finch brain, suggesting that *RPS6* promoter is able to integrate both cell-autonomous, sex-specific transcriptional control and sex hormone-mediated systemic regulation. This latter aspect was supported by the presence of estrogen receptor binding sites in avian *RPS6* promoters (Fig. 2A). It may also be a conserved feature in mammals, as *RPS6* was recently identified as a significant indicator for sex-specific prognosis of diffuse glioma in humans [49].

## Conclusions

In this work, we used embryonic fibroblast cells from seven birds and showed that avian chrZ genes were highly dosage-compensated, similar in its extent to mammalian chrX. Avian and mammalian dosage compensations likely involve different molecular mechanisms. Some chrZ genes were not compensated and may play a role in sex determination. *RPS6* was identified as the only chrZ gene exhibiting male-biased expression in all species examined.

## Methods

### Egg collection and fibroblast cell preparation

Avian eggs from seven different species [chicken (*Gallus gallus domesticus* ), quail (*Coturnix coturnix japonica*), turkey (*Meleagris gallopavo domesticus*), blue peafowl (*Pavo cristatus*), duck (*Anas platyrhynchos domesticus*), zebra finch (*Taeniopygia guttata*) and emu (*Dromaius novaehollandiae*)] were obtained from the following sources in Japan: chicken eggs from Shimojima Farm, Kanagawa; blue peafowl eggs from Kumamoto City Zoo and Botanical Gardens, Kumamoto; quail eggs from Motoki corporation, Saitama; duck eggs from Shiina Hatchery Co., Ltd. , Chiba; turkey eggs from Shimanto Turkey Growers Association, Kochi; emu eggs from Japan Eco System Co., Ltd., Fukuoka; and zebra finch eggs from Hokkaido University, Hokkaido. For fibroblast cell preparation, chicken eggs were incubated at 38.5 °C to reach to stage HH36 [50]. Developmental speed and total incubation time vary among birds [51]. Embryos of all other species were incubated at 37.5–38.5 °C to reach the equivalent stage of chicken HH36. This corresponded approximately to day 10 (D10) in chicken, D9 in quail, D13

in duck and blue peafowl, D14 in turkey, D23 in emu, and D8 in zebra finch. Embryos were then treated by removal of head and neck, limb, skin and fat deposit, and internal organs. The remained trunk portion was trypsinized at 37 °C, followed by neutralization and centrifugation to obtain digested cell pellet. Cell pellet was resuspended with DMEM/10%FBS (FBS Cat# S1820-500 from Biowest, France; DMEM high glucose Cat# 043-30085 from FUJIFILM Wako Chemicals, Japan) and seeded in DMEM/10%FBS or DMEM/F12/10%FBS (DMEM/F12 Cat#042-30795 from FUJIFILM Wako Chemicals, Japan) for expansion.

### CAGE-seq

Total RNA was prepared using commercial reagents (TRIzol, Invitrogen) from each sample of avian embryonic fibroblast cells at the fifth Passage. Five micrograms of RNA from each fibroblast sample was used for nAnTi-CAGE library preparation and subsequent sequencing using the Illumina HiSeq2500 platform (K. K. DNA-Form, Yokohama, Japan). Quality of sequenced CAGE reads was checked by using FastQC v0.11.5 and then trimmed with fastx_trimmer -Q33 -l 75 (FASTX Toolkit 0.0.14). Moirai removeN script was applied to remove reads with "N" nucleotides. CAGE reads matching to adapters or rRNA (Moirai defined rRNA for gal-Gal3) were removed with Trimmomatic-0.38 and RNAdust 1.06, respectively. Trimmed CAGE reads were aligned with BWA 0.7.10-r789 and unmapped reads realigned with Hisat2 v2.1.0 to following reference assemblies: chicken - galGal6 (GCF_000002315.6), quail - Coturnix_japonica_2.1 (GCF_001577835.2), turkey - melGal5 (GCF_000146605.3), turkey - MGAL_WU_HG_1.0 (GCA_905368555.1), peacock - GT_SO_6221 (http://gigadb.org/dataset/100559), duck - ZJU1.0 (GCF_015476345.1), zebra finch - bTaeGut2.pat.W.v2 (GCF_008822105.2), emu - droNov1 (GCF_003342905.1), emu - ZJU1.0 (GCA_016128335.1). Alignments were processed into CTSS (CAGE transcription start sites) and CAGE peaks with PromoterPipeline [52] with default threshold of at least 10 TPM (tags per million) in one of the samples. CTSS and CAGE peaks data were deposited in Zenbu (https://fantom.gsc.riken.jp/zenbu/reports/#Birds_Promoter_Atlas). CAGE libraries of all bird species and Chicken FANTOM data [28] were realigned to galGal6 and analyzed using the same approach. CAGE peaks were associated with nearby transcripts located on the same strand by using ChIPseeker v1.32.0 package for R. Gene models for MGAL_WU_HG_1.0 were built by using Augustus v3.2.3 with --species=chicken option and hints based on EST and mRNA for melGal5 obtained from UCSC and mapped with blat v.36x2 to MGAL_WU_HG_1.0. Completeness of gene annotation accessed by BUSCO v3.0.2 (76.8% complete). 2kb regions of CAGE peaks located within 100 bp from transcript start were used as a training set for TSSClassifier [24], and remaining distal intergenic, intronic, exonic, and UTR CAGE peaks were analyzed and classified into "promoters" or "not-promoters" TSS peaks. For the rest of the analysis, we used only promoter region localized (within 3 kb) peaks or other TSS peaks classified as "promoter." Differential expression analysis was carried by using edgeR v3.38.1 package for R for each species separately on TPM counts using Generalized linear models' approach. Cross-species comparison was carried out for the purpose of validation (Additional file 1: Fig. S13) through

previously described approach [53] which uses modified variance-stabilizing transformation. Human dermal and mouse embryonic fibroblast samples were obtained from FANTOM6 and 5 repositories, respectively. Orthologs defined by using blastp (v2.2.29) -evalue 1e-5 against chicken protein sequences. Species specific genes were obtained by applying differential expression approach on cross-species normalized counts. To note, peacock genome is assembled to the level of scaffolds and chromosomes information is not available. For Additional file 1: Fig. S5, chicken sex chromosomes were used for Blastn against peacock assembly and the best match scaffold considered as a sex chromosome in peacock. Quail, peacock, and turkey genome assemblies do not include chrW.

### RNA-seq

Paired end libraries for RNAseq were prepared for each species using 1 female and 1 male samples from each species (a subset of samples used for CAGE-seq) and sequenced on Illumina HiSeq2500 (K. K. DNAForm, Yokohama, Japan). Reads were trimmed against adapter sequences and rRNA with Trimmomatic-0.38, aligned to the genome assemblies with Hisat2 v2.1.0, and counted against gene models with HTSeq v2.0.1. FPKM values were counted with edgeR package for R. Batch effect correction for MDS plot was done by using limma ("removeBatchEffect" function). scRNAseq analysis of published HH4-HH7 and S4-S13 data [54, 55] was carried out by using cellranger-7.0.1 and galGal6 as a reference. Obtained counts were processed through Pagoda2 and Conos packages with default settings to make integrated normalized expression matrix.

### qPCR

Total RNA extracted from fibroblast cells was used to obtain cDNA with SuperScript III Reverse Transcriptase kit (Thermo Fisher). Primers were designed for *RPS6* as follows: forward primer #1 matching exon 2 and 3 GGAGTGGAAGGGCTATGTTG, reverse primer #1 matching exon 4 – TTGAACAGCTTGCGGAT, insert size 309 bp; forward primer #2 matching exon 3 GACGTGTCCGCCTTCTGCTC; reverse primer #2 matching exon 4 – TTCCTCACAACATACTGGCG, insert size 268 bp. Two genes with stable expression profiles *ARPP19* and *YWHAE* were selected for control with primers as follows: *ARPP19* forward primer on exon 2 – TGAAAGCAA GATACCCTCAT, reverse on exon 3 – TCTTCATCTTTGCTTTAGCC, size 126 bp; *YWHAE* forward on exon 2 – CAGTGGAAGAAAGAAACCTG, reverse on exon 3 – GAATGAGGTGTTTGTCCAGT, size 216 bp. Efficiency of primers was tested in series of cDNA dilutions 1:10, 1:100, 1:1000, 1:10000, and *RPS6* primers #2 with *YWHAE* as control were selected for qPCR. The primer efficiency calculated as $E = (10^{(-1/Slope)}-1) \times 100$ (Additional file 1: Fig. S11B). qPCR was done on Light-Cycler 96 (Roche) with at least two technical and two biological replicates. Data was analyzed in pcr v1.2.2 package for R. Nucleotide alignment for *RPS6* CDS and tree (Neighbor Joining method, Jukes-Cantor distance measure, bootstrap 1k) was created by using CLC Genomic Workbench 20.0.

## Supplementary Information

---

**Additional file 1: Fig. S1.** Experimental design and the data analysis pipeline. **Fig. S2.** Correlation between CAGE replicates. **Fig. S3.** Correlation between CAGE-seq and RNA-seq methods in birds. **Fig. S4.** Peak distribution and model efficiency estimation. **Fig. S5.** Sample MDS plots and hierarchical clustering. **Fig. S6.** Expression profiles for gametologue genes defined by blast between chrW and chrZ genes. **Fig. S7.** Genomic views. **Fig. S8.** Autocorrelation of CAGE expression on chromosome. **Fig. S9.** Top male-female marker genes in avian fibroblast cells. **Fig. S10.** RPS6 is a universal male-biased marker. **Fig. S11.** RPS6 expression in chicken embryonic cells and qPCR validation. **Fig. S12.** Medium composition effect contribution on sex chromosome gene expression in fibroblast cells. **Fig. S13.** Cross-species comparison of fibroblast gene expression.

**Additional file 2: Table S1.** Embryonic fibroblast samples statistics.

**Additional file 3: Table S2.** Transcribed promoters identification in birds fibroblasts.

**Additional file 4: Table S3.** Numbers of differentially expressed genes between males and females samples

**Additional file 5: Table S4.** Sex chromosome gene expression contribution to sex differences of fibroblast cultures

**Additional file 6: Table S5.** Dimorphic gene enrichment per chromosome (Fisher exact test for DE genes FDR < 0.05 vs expressed genes)

**Additional file 7: Table S6.** Gametologue genes impact on differential expression estimates between male and female fibroblasts

**Additional file 8: Table S7.** CAGE peaks annotation and differential expression analysis (male vs female)

**Additional file 9: Table S8.** Effect of medium on RPS6 expression. Differential expression analysis for DMEM/10%FBS vs DMEM/F12/10%FBS

**Additional file 10.** Review history

---

### Acknowledgements

We would like to thank K. Matsushita from Shimanto Turkey Growers Association in Kochi, Japan, for supplying us with turkey eggs.

### Review history

The review history is available as Additional file 10.

### Peer review information

### Authors' contributions

Conceptualization: GS, OG, RD, and HK; methodology: GS, OG, RD, HK, WS, and S. Intarapat; formal analysis: GS, OG, RD, HK, GI, AS, WS, and S. Intarapat; resources: GS, HK, GI, KW, S. Ide, NT, HZ, WS, and S. Intarapat; writing—original draft: GS, RD, OG, HK; writing—review and editing: GS, RD, OG, HK, WS, S. Intarapat, and KW.

### Funding

This work was supported by Japan Science and Technology Agency (JST) e-ASIA joint research project grant JPMJSC19E5 and by a Biotechnology for Bird Conservation grant from Revive & Restore during manuscript revision. RD, AS, and OG were supported by Ministry of Science and Higher Education of the Russian Federation grant 075-15-2021-1344.

### Availability of data and materials

All transcriptomics data created in this work is deposited in GEO NCBI (GSE213253) as raw and processed expression tables with gene annotations [56]. Genomic views are available through Zenbu-reports [57] browser https://fantom.gsc.riken.jp/zenbu/reports/#Birds_Promoter_Atlas. Single cell RNAseq data for chicken embryo were accessed through GSE181577 [58] and GSE223189 [59].

## Declarations

### Ethics approval and consent to participate

This research does not involve human or adult animal subjects. Avian embryonic fibroblast collection was performed following the biosafety and bioethical regulations of the Kumamoto University.

### Competing interests

The authors declare no competing interests.

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

## 
