## [**Additional file 10.** Review history · Genome Biology]

Review History

First round of review

Reviewer 1

Are you able to assess all statistics in the manuscript, including the appropriateness of statistical tests used? Yes, and I have assessed the statistics in my report.

Comments to author:

Deviatiarov, et al. here presented a report titled as "Dosage compensation of Z sex chromosome genes in avian fibroblast cells". This is a work focusing on the sex dimorphic expressed genes of fibroblast cells across 7 avian species by CAGE sequencing and reached a new conclusion that about 3/4 of chrZ genes are strictly compensated across Aves, similar to mammalian chrX. The results is very interesting, and here I raise my main comments regarding some technical aspects of the work, that remain for the authors to clarify. It is important to elucidate why the conclusion is different with the previous views on dosage compensation in birds based on the mRNA-seq. And I expect minor revisions of the work from the authors.

Major Points:

1. CAGE-seq is used for quantitative transcriptome analysis on TSS. Why do the authors think it is more advantageous than traditional mRNA-seq for quantifying the expression level for studying the dosage compensation of birds? This needs to be explained in details in the introduction part. Although I did notice a brief introduction of this at line 67. What work has been done with CAGE-seq in other species, that have produced a different conclusion with mRNA-seq in birds or other species? Does CAGE-seq generally show a good or bad correlation with mRNA-seq?
2. It is quite surprising that quail, with a highly differentiated sex chromosomes, actually have 94% of the chrZ-linked genes with inferred dosage compensation. Are there any explanations for the quail case? And for emu, did the authors use all the genes on the chrZ, or only referring to genes in the non-PAR region. Because it is quite expected that genes in PAR would have an equal expression level between sexes. And how is this percentage, 70% of chicken, and 75% of zinch, compared to previously published results in chicken and zebra finch?
3. 'On average, 8582 TSS peaks representing 7608 genes' in line 74. This number seems to be a bit low. What is the number of expressed genes in mRNA-seq of fibroblast cell among one of these species?
4. Line 83-84: How exactly did the authors get this number of 73.4%?
5. Line 116, DMRT1 is not expressed in fibroblast cells, this is likely expected as it specifically functions in the gonad. But this is, in my understanding, not the definition of cell autosomes sex identity. As for many other species whose sex determination is not cell autonomous, DMRT1 likely also does not express in the somatic cells. Please rephrase.

6. Line 112: "whereas compensated genes clustered to the PAR (pseudo-autosomal region) (Sup-Fig.6A)". How are compensated genes clustered in the PAR? Because PAR has an equal copies of genes between sexes, no dosage compensation is expected there.

7. I am also curious that why the authors specifically listed AHCY, HSPA8 etc. in the text, any previous studies on these genes.

Minor Points:

8. Line 81, 'Sup-Figu4' to 'Sup-Fig.4'

9. Line 93, specifies these 'dosage-compensated genes' are all located at the non-recombination regions, same for Line 98.

10. Line 101, as known avians lack global dosage compensation, maybe should change the words 'as effective as' to 'possibly carry a similar proportion of dosage compensation genes as mammals'.

11. Line 113, 'ratites' to 'avian', as SO and PAR are features not restricted to ratites.

12. Line 117, DMTR1 is a putative sex determination factor.

13. Line 166, should explain what the 'MDS' represent.

14. Line 169, expression of sex marker genes of which species? Same for Line 171.

15. Line 226, Sup.Fig6 is not very clear for the CAGE track.

16. Line 240, words were covered by icons in the phylogenetic distances figure.

17. Line 288, a space between 'carriedfor'

Reviewer 2

Are you able to assess all statistics in the manuscript, including the appropriateness of statistical tests used? No, I do not feel adequately qualified to assess the statistics.

Comments to author:

Deviatiarov et al. provide a comprehensive evaluation of sex differential gene expression in primary fibroblastic cells cultures from a wide variety of birds. The results provided in this research result of great interest for the bird sex determination community and introduce the identification of new candidates for cell autonomous sex identity of cells, which remains unknown in non-gonadal tissues. The identification of the Z-linked ribosomal gene RPS6 highly expressed in males of all evaluated species provides a new interesting candidate for this theory.

Unfortunately, there are some issues that should be addressed before this research is ready for publication.

Major comments:

- * Please provide more details on the media used. Does the media used contain phenol red or not? Does the serum contained hormones, or it was charcoal stripped to eliminate androgens and estrogens? This should be discussed, in particular as RSP6 seems to have some estrogen responsive elements and can be affected by the presence of estrogen in the media, which is absent in developing male embryos. Is RSP6 differentially expressed in male and female bird embryos / recently isolated fibroblasts?
- * The effect of different media on the cultures was never discussed. It seems that the media used had a major effect in females than males. In supplementary figure 3B female samples cluster together by media first and then by sex, whereas in the males, this does not seem to be the case. Including a different feature (like different shape) in supplementary figure 3A with the media used could be important to evaluate the contribution of the different media to the different dimensions in the PCA plots.
- * Despite being in different sex chromosomes, gametologue pairs could have an additive effect resulting in similar combined levels of expression in ZZ males and females ZW. Do you see a similar effect in the fibroblast samples? How does the number of differentially expressed genes between male and female samples change when you consider gametologue expression?
- * As this is the main focus of the paper, more information should be provided regarding the differentially expressed genes between male and female fibroblasts but shared between species. For example, an upset plot could be generated for both male and female DEG shared between species. This could be also accompanied with a supplementary table with gene lists shared among the different species.
- * RSP6 is mentioned to be the only chrZ gene exhibiting male-biased expression in all species examined, but in figure 2C, both quail and duck show non-significantly different expression between male and female samples ($p > 0.05$).
- * Please increment the sample number of the qPCR data, as two biological replicates is not enough to perform statistics. Please provide the number of biological replicates for each specie/sex.

Minor comments:

- * Line 57, gender is a social construct only attributed to humans and not birds. Please replace it with sex.
- * Although the prediction power of the sex specific genes in different mixed samples is interesting, supplementary figure 4 does not provide any important information to the manuscript story and can be removed.
- * Supplementary figure 6 is referenced in the text before supplementary figure 5.
- * In supplementary figure 3B, the sex of the samples is not correctly referenced in turkey, peacock and zebrafish.
- * Supplementary figure 8 is referenced in the text (methods) before supplementary figure 7b.
- * Supplementary figure 8 is only discussed in the methodology. I suggest moving this figure to the beginning of the manuscript (may be to be sup figure 2), as it involves the cross-species comparison and provides information regarding the sequenced samples.

* As the developmental stage in some birds can include different incubation days, it would be precise to include the days/hours of incubation of all the different birds until collection. This could be included in a table format. This could be useful to ensure reproducibility.

Reviewer 3

Are you able to assess all statistics in the manuscript, including the appropriateness of statistical tests used? Yes, and I have assessed the statistics in my report.

Comments to author:

The authors harvested and cultured fibroblast cells from six bird species (chicken, quail, turkey, blue peafowl, duck, zebra finch and emu), having two to four replicates per sex. They assess gene expression levels using CAGE-seq, a method that preferentially retrieves transcriptions start sites of expressed genes and which is typically used for promoter characterization. To my understanding the authors used CAGE-seq to interrogate sex-biased gene expression but do not present an in depth analysis on promoter recovery and characterization that is enabled by CAGE-seq. They do mention the genomic distribution of CAGE peaks but do not show an association to gene expression (the authors did not produce complementary RNA-seq data). The manuscript presents proportions of compensated and non-compensated genes on the Z across the six bird species. One of their main conclusions is that fibroblast cells show high dosage compensation levels, comparable to what has been observed in mammals (~77%), a result that by itself it is interesting. Taking advantage of their phylogenetic sampling, they could also report the proportion of shared dosage compensated genes across birds. Furthermore, they report sex-biased gene expression on the Z and suggest how these genes could play a role in sex-determination, but fail into showing a link between the two variables. Finally, they found that the gene RPS6 shows a conserved male-biased gene expression across the investigated birds.

Comments to the authors:

1. The introduction as it is at the moment, fails to communicate the reader what are the objectives of the study and what is the approach the authors took to fill the knowledge gap regarding sex chromosome dosage compensation in birds. Two to three sentences summarizing the main findings of the work are also missing.
2. The first paragraph of the results section mostly describes methods and do not describe obtained results. Thus, this part should be transfer to the methods section.
3. The authors produced CAGE-seq data to recover TSS of expressed genes and annotated CAGE peaks using genome annotation information, assuming that the most nearby gene from the CAGE peak is expressed. How well does CAGE-seq and RNA-seq correlate? This seems to be an important point since significant peaks can be false-positives.
4. Line 78: How was statistically tested that genes expressed in the W chromosome had the strongest clustering effect between the samples? How many genes on the W were retrieved that showed sex-biased expression in each species? The differential clustering observed in Figure 1D could also come from sex-biased genes located in the autosomes or the Z chromosome. If the authors want to state that genes expressed on the W have the highest sample clustering effect, this should be statistically tested.

5. Why are there only two technical replicates for the male chicken data and double the amount for females? What are the correlation levels of within sex replicates? Could the authors hypothesize why female data shows a higher variance when compared to male replicates?
6. Figure 1G is not clear to me. From the main text, it describes that the figure shows male/female ratios on the autosomes, Z and W chromosomes. The figure plots, show male and female log₂FC boxplot distribution from the three chromosome types. How was log₂FC calculated, shown in the male and female labeled panel of figure 1G?
7. The authors suggest that sex-biased expression of chrZ/chrW genes possible control sex determination, and that the autosomes have an influence through their own sex-biased expressed genes. What is the data presented connecting the effects of sex-biased gene expression in the autosomes to the expression of genes in the Z?
8. Line 87: How did the authors assess sex-specific expression?
9. Did the authors check how many expressed genes on the Z are on the pseudo autosomal region (PAR)? It seems that this was only checked in the emu. The quail data shows that 94% of the genes are compensated. How big is the quail's PAR?
10. Line 110: How was the lack of regionalized distribution of compensated and non-compensated genes tested? Sup-Fig. 6A does not show data depicting the distribution of compensated and non-compensated genes on the Z chromosome. An autocorrelation analysis could be done along the Z chromosome to uncover enrichment or depletion of specific gene expression categories.
11. Line 116: Different levels of sex-biased gene expression are observed throughout the development of different organisms. A particular set of genes will probably show sex-biased gene expression even after gonadal differentiation.
12. In the last part of the manuscript the authors spend some time on the cell autonomous sex identity (CASI) phenomenon. However, CASI is not mention in the introduction and there is no mention on why looking at CASI is important.
13. What is the link of loc112530614 expression to the cell autonomous sex identity (CASI) phenomenon? The authors do not mention if the expression of loc112530614 is conserved in each of the bird species that they analyzed. Similarly, it is not clear what is the role of HINTW expression to CASI. The authors only describe its expression in the bird species studied but fail to state a clear link between HINTW expression and CASI.
14. Line 130: How do these novel genes contribute to the understanding of CASI. Why are AHCY, HSPA8, COL4A1 and RAD23 genes important?
15. The conclusions are very vague and do not link the study's results to the contribution of dosage compensation mechanisms in birds. One of their main conclusions is that non-compensated genes on the Z play a role in sex-determination, but they fail to show a link between sex-biased gene expression and sex-determination.

Minor comments:

Line 31: I would write "The molecular mechanisms regulating avian sex determination are not well-understood".

Line 37: "Suppression of meiotic recombination of ChrZ/chrX led to their degeneration into chrW/chrY, respectively". Citation is missing.

Line 48: Replace "sex dimorphism" with "sexual dimorphism".

Line 53: This sentence needs to be rephrased.

Line 57: The supplementary text file was missing from the documents to be downloaded.

Lines 53-64: This paragraph is mainly describing methods and not results.

Line 70: "sex dimorphic gene expression" replace with sex-biased gene expression.
Line 83-84: What do the authors mean by sex specification? Do they mean sex-determination?
The message of this sentences is not clear to me. Are the authors suggesting that sex-biased expression in the autosomes have an impact on sex-biased expression on the sex-chromosomes?
Line 121: Sup-Fig.5 should be mentioned before Sup-Fig.6
Line 126: Before "Second", citation is missing.
Line 140: In "suggesting that its involvement" delete the "that".
Line 164: The vertebrate phylogeny is also shown, together with the avian phylogeny.
Figure 1B, does the figure show total of annotated genes or total of expressed genes, together with their corresponding TSS?
Figure 1F, For which bird species is the data presented in this panel?
Figure 1G, which statistical test was used to assess significance? From which bird species is the data presented? Are genes on the PAR included in the boxplot representing the Z chromosome?
Figure 1H, chromosome labels are not in the correct order. Did the authors performed a chromosome enrichment analysis?

Authors Response

Point-by-point responses to the reviewers' comments:

Comment

Response

Second round of review

Reviewer 1

The author has addressed my comments raised from the last round of revision with satisfactory.

Reviewer 2

The reviewers provided the required revisions. As the authors mentioned, it is on the editorial side to decide if figures in the response to the authors should be included in the manuscript or not.

Reviewer 3

In my opinion, the authors took great care in answering all the comments in the response letter. Nevertheless, when reading the manuscript, there are still some of the issues that were raised in the first round of revisions unresolved. Bellow, were are some comments regarding the manuscript.

Line 37. "ChrZ" is capitalized.

Line 39. The use of "sex dimorphism" is confusing in this sentence. I guess the authors refer to sex determination.

Line 43. This statement is only true for placental mammals, in *Drosophila* for example, full dosage compensation is achieved without XIST.

Line 51. The Introduction does still feel a little bit chopped. It lacks a description to the readers on the main objectives of the manuscript, explaining what the authors want to find out in their study.

Line 78. A sentence should not begin with a number, in this case, 4.5%.

Line 82. There is a full stop after “Sup-Fig.5B”.

Line 83. Do the expression “dimorphic genes”, refer to the CAGE or the gene expression data? This is not entirely clear in this sentence.

Line 92. Are the authors presenting ratios or percentages?

Line 94. How did the exclusion of the peafowl data affected the retrieved percentages for the rest of the species?

Line 113. I might have missed this information, but which methods were used to assess enrichment of dosage- and non-dosage compensated genes along the Z? Was the analysis mentioned in this line, independent from the autocorrelation study?

Line 128. Why is CASI important in the context of dosage compensation?

Line 130. Genes on the MHM, have a tendency of showing lower gene expression in males, thus giving a f:m expression ratio close to the expectation of expression compensation. Only in certain developmental stages there is gene expression inactivation a the MHM, but not in all (<https://doi.org/10.1534/genetics.115.179234>).

Line 134. If HINTW is located in the no PAR region of the W, it is expected to show female biased expression. There is still missing mentioning the new insights that the expression pattern of LOC112530614 and HINTW, bring to have a better understanding of the phenomenon CASI.

Line 37. "ChrZ" is capitalized.

>Modified as recommended.

Line 39. The use of "sex dimorphism" is confusing in this sentence. I guess the authors refer to sex determination.

>Modified as recommended. This sentence now reads: "..., sex determination is governed not genetically, but rather via environmental factors (e.g., temperature) during critical developmental windows"

Line 43. This statement is only true for placental mammals, in Drosophila for example, full dosage compensation is achieved without XIST.

>We agree. To avoid un-intended confusion, we have modified the sentence as below: "A well-studied example of dosage-compensation is XIST (a long non-coding RNA)-mediated chrX condensation and subsequent inactivation in placental mammals"

Line 51. The Introduction does still feel a little bit chopped. It lacks a description to the readers on the main objectives of the manuscript, explaining what the authors want to find out in their study.

>We adjusted the last sentence of the introduction for clarity. We feel that the objective of this work is mentioned clearly in the first sentence of the result section, so we do not want to repeat it unnecessarily. The manuscript now reads (the last sentence of intro and the first sentence of result): "In addition to dimorphism in gonadal morphology, birds exhibit secondary sexual dimorphism (e.g., body weight, plumage shape, wattle size and courtship behavior) which is mainly under cell-autonomous regulation via inherited genetic identity and weakly modulated by sex hormones after gonadal differentiation. **Results and Discussion** To investigate how sex-specific differentiation in birds is programmed cell-autonomously, we compared transcriptional activities in purified male and female embryonic fibroblast cells from seven avian species (the chicken, quail, turkey, blue peafowl, duck, zebra finch and emu),....."

Line 78. A sentence should not begin with a number, in this case, 4.5%.

>We modified the sentence to avoid this.

Line 82. There is a full stop after "Sup-Fig.5B".

>Corrected

Line 83. Do the expression "dimorphic genes", refer to the CAGE or the gene expression data? This is not entirely clear in this sentence.

> We replaced "dimorphic genes" with "differentially expressed genes".

Line 92. Are the authors presenting ratios or percentages?

>We have rephrase this sentence as below: modified the sentence to avoid this.

Line 94. How did the exclusion of the peafowl data affected the retrieved percentages for the rest of the species?

> We thank reviewer for the comment. If we keep the peacock data this number will become irrelevant due to absence of assembled sex chromosomes. We stated this point in the sentence. No changes required here.

Line 113. I might have missed this information, but which methods were used to assess enrichment of dosage- and non-dosage compensated genes along the Z? Was the analysis mentioned in this line, independent from the autocorrelation study?

>We thank reviewer for the comment. The conclusion here is based on autocorrelation analysis. In order to avoid misunderstanding we rephrased the sentence: "Our data indicated obvious regionalized distribution of either compensated or non-compensated genes on avian chrZ in the case of emu, in which male-biased chrZ genes appeared to cluster to the stratum 0, whereas PAR (pseudo-autosomal region)-located genes, as expected, did not show sex-specific expression shifts (Additional file 1: Fig. S7A)."

Line 128. Why is CASI important in the context of dosage compensation?

>We thank reviewer for the comment. Non-compensated genes are presumably involved in sexual differentiation and sexual dimorphism, constituting molecular background (landscape) of the CASI. We

adjusted this sentence as follow: “We noticed four interesting features, both validating previous findings and offering new insight to dosage compensation and CASI.”

Line 130. Genes on the MHM, have a tendency of showing lower gene expression in males, thus giving a f:m expression ratio close to the expectation of expression compensation. Only in certain developmental stages there is gene expression inactivation a the MHM, but not in all (<https://doi.org/10.1534/genetics.115.179234>).

>We thank reviewer for the comment. We agree with the reviewer and have added the reference and rephrased the sentences as below: “**First**, one chicken chrZ gene (LOC112530614) exhibited strong inactivation in male fibroblast cells. This gene is located within the *MHM1* (male hypermethylation 1) locus (**Additional file 1: Fig. S7B**) reported to be downregulated in male and active in female [14, 17, 40] and hypothesized to positively regulate female chrZ genes [41]. Although this *MHM1* locus is conserved in Galloanserae, it is not universally present in birds [14, 42].”

Line 134. If HINTW is located in the no PAR region of the W, it is expected to show female biased expression. There is still missing mentioning the new insights that the expression pattern of LOC112530614 and HINTW, bring to have a better understanding of the phenomenon CASI.

>We thank reviewer for the comment. We think strong and universal female-biased expression of HINTW underscores its potential role in sexual differentiation and CASI, similar to RPS6 for male cells. Unlike RPS6, however, annotation of HINTW gene relies on chrW assembly and therefore its universal expression remains to be confirmed in species like emu. For LOC11250614, we specified in this work active TSSs in the MHM locus of chicken chrZ (and in particular the ncRNA gene LOC112530614), making it possible to study genomic regulatory elements in MHM and transcriptional regulation of LOC112530614.